# Health Education Intervention on Hearing Health Risk Behaviors in College Students

**DOI:** 10.3390/ijerph18041560

**Published:** 2021-02-06

**Authors:** Dahui Wang, Chenhui Li, Yi Wang, Shichang Wang, Shuang Wu, Shiyan Zhang, Liangwen Xu

**Affiliations:** Department of Medicine, Hangzhou Normal University, Hangzhou 311121, China; 20120051@hznu.edu.cn (D.W.); 2018111012012@stu.hznu.edu.cn (C.L.); wangyi2@stu.hznu.edu.cn (Y.W.); 2018111012039@stu.hznu.edu.cn (S.W.); wushuang1995@stu.hznu.edu.cn (S.W.); 2020011012006@stu.hznu.edu.cn (S.Z.)

**Keywords:** college students, hearing loss, behavior, health education, health belief model

## Abstract

Young people, like college students, are at risk of hearing loss from prolonged and excessive exposure to loud sounds. However, behavioral interventional studies on them are inadequate. This study explored the application of a health belief model to the health education intervention on college students for improving hearing health knowledge, health belief, and hearing behaviors. From November 2017 to September 2018, a cluster randomized controlled trial was conducted, enrolling 830 college students, with 419 in the intervention group and 411 in the control group. The intervention group received a 3-month hearing health education, while the control group received no intervention. The information of hearing health knowledge, health belief, and hearing behaviors were collected using hearing health questionnaires before the intervention, after the intervention, and 3 months after the intervention cessation. The intervention significantly improved hearing health knowledge, health belief, perceived severity, and self-efficacy in female students, and effectively reduced the frequency of using headphones per day, duration of using headphones each time, and proportion of using headphones at high volume in female students, and reduced the behaviors of sleeping with headphones listening in females and males. Therefore, this study confirms the effectiveness of health belief model-based intervention for changing hearing loss-related risk behaviors.

## 1. Introduction

In the 2015 Global Burden of Disease Study, hearing loss was ranked as the fourth most prevalent chronic disease worldwide [1]. The World Health Organization (WHO) reported that globally 1.1 billion young people were at risk of hearing loss due to prolonged and excessive exposure to loud sounds [2]. As young people, college students have always been a big concern, especially considering the current situation of hearing health-related knowledge, beliefs, and risk behaviors.

Tung et al. found that 11.9% of 1878 first-year college students in a university in Taiwan had unilateral or bilateral hearing thresholds above 25 dB [3]. Balanay et al. found that 39.6% of 2151 college students reported at least one hearing symptom, with ear pain being the most common [4]. Studies have shown that the main noise sources among college students are a variety of recreational noise activities, such as attending concerts, bars, karaoke sessions, and especially portable music players [4,5,6]. Headphones are convenient and improve the auditory sound experience but also increase the risk of hearing loss [7,8]. Sun et al. found that 66.2% of 1009 Korean college students used earplugs to listen to their portable music players [7]. Compared with common earplugs, the preferred listening level of noise-canceling earphones in the presence of background noise can be reduced by 4 dB at the most [8].

Since there is no effective treatment for hearing loss, it is worth advocating for an active intervention to reduce the risk factors and incidence of hearing loss through health education [5]. Kathleen et al. found that most students were willing to change their behavior, like shortening the duration of personal hearing equipment use or reducing their listening volume level if information from an audiologist or doctor increased their awareness regarding unsafe hearing behavior [9].

Based on the principles of cognitive theory, the health belief model holds that expectations and beliefs play a key role in persuading people to change harmful behaviors [10]. Hearing health education programs contribute to the promotion of hearing health knowledge, as well as the development and increase in hearing health protection beliefs and behaviors [11]. Knobel et al. conducted a 3-month hearing loss classroom education on 271 students in grades three to five. The knowledge, attitude, and intentional behaviors related to the prevention of noise-induced hearing impairment were significantly improved [12]. A study by Keppler et al. found that, through an effective 6-month hearing education program, 78 participants showed improved scores regarding their attitudes toward noise and their beliefs toward hearing protection and hearing loss, with 12% participants using hearing protectors more frequently [13].

Studies are still lacking on studies performed on hearing health-related behavioral interventions in college students. This study aimed to examine the effects of a health intervention on hearing health risk behaviors among college students to provide new ideas for the design and application of a hearing health intervention scheme in universities.

## 2. Materials and Methods 

### 2.1. Participants

Classes were cluster-randomized sampled from November 2017 to September 2018, and a total of 830 eligible college students were enrolled from the Department of Medicine, Hangzhou Normal University, with 419 students in the intervention group (IG) and 411 students in the control group (CG). The inclusion criteria were as follows: (1) medical students living on the university campus; (2) signed the informed consent and agreed to participate in the intervention program; and (3) without hearing loss. The hearing status of these college students was tested in our previous research [14].

### 2.2. Hearing Health Intervention

The intervention group was provided with online and offline hearing health education. The online intervention included releasing hearing health knowledge through the public account “Medical Ear” of WeChat (Tencent, Shenzhen, China) and establishing WeChat groups for group communication. Offline intervention predominantly comprised the distribution of hearing health leaflets, teaching ear protection exercises, and holding hearing health knowledge competitions. The control group did not receive any intervention (Figure A1).

### 2.3. Questionnaire Survey

The questionnaire was designed to collect the demographic variables, hearing health knowledge, hearing health beliefs, and hearing loss-related behaviors.

The demographic variables include age, grades, ear problems (tinnitus, earache, and ear tightness), and the number of people with hearing loss in the family. Tinnitus was divided into two conditions: no tinnitus within the past year, and occasional tinnitus (less than or equal to once a week), or frequent tinnitus (more than or equal to twice a week). Earache and ear tightness were similarly classified.

The hearing health knowledge dimension contained 36 items, including hearing risk factors (11 questions), harms of noise to the hearing system (six questions), early symptoms of hearing loss (eight questions), and methods for hearing loss prevention (11 questions). Every correct answer was awarded 1 point, whereas the wrong answers or ‘do not know’ answers were awarded 0 point.

The hearing health belief dimension had 45 items, including perceived susceptibility to hearing loss (five questions), perceived severity for hearing loss (nine questions), perceived benefits (seven questions), perceived barriers to changing harmful behavior for hearing-loss (eight questions), cues to action (eight questions), and self-efficacy (eight questions). A 5-point Likert scale rating was used, and a value was assigned from “totally disagree” to “completely agree” with total scores being 45–225 points. The higher the score, the better the level of belief in hearing health. The questionnaire was validated in previous research with 2088 participants involved. Its Cronbach’s coefficient was 0.888, and the KMO value was 0.872. A principal component analysis showed that confirmatory factors accounted for 64.395% of the total variance. 

We surveyed six types of hearing health-related behaviors: (1) average frequency of using headphones ≥3 times/day; (2) average duration of using headphones ≥30 min each time; (3) probability of increasing headphone volume in a noisy environment ≥50%; (4) volume of the headphones ≥40% of the total volume: (5) using in-ear type headphones; (6) sleeping with headphones for listening to music or radio.

Questionnaires were administered to both groups of students before the intervention (T1), at the intervention cessation phase (T2), and 3 months after the intervention cessation (maintenance phase; (T3). The questionnaire survey was conducted by investigators, including teachers and graduate students who received strictly professional training to ensure data collection quality. They were able to provide accurate interpretation during questionnaire collection by face to face interviews in order to minimize the subjective bias of the participants.

### 2.4. Statistical Analysis

The database was established using Epidata 3.0, and analyses were performed using SPSS 22.0 for Windows (IBM Corp., Armonk, NY, USA). The demographic data for each student were compiled to produce descriptive statistics. Continuous variables were shown as the mean ± SD (Standard Deviation), and categorical variables showed as *n* (%). The differences among groups at baseline were analyzed with *t*-test and chi-square test. The change in scores of health knowledge and health beliefs were calculated as the value obtained at T2 and T3 minus the baseline value, and Repeated Measures ANOVA (Analysis of Variance) were used to analyze the differences in intervention outcomes between the control and intervention groups. If one type of behavior was found to have improved at T2 or T3, compared to T1, the intervention was considered effective (Y = 1), whereas if there was no improvement or the behaviors were found to have worsened, the intervention was considered ineffective (Y = 0). Binary logistic regression analysis was used to estimate the efficiency of health interventions on the hearing of health-related behaviors, and *p* < 0.05 was considered statistically significant.

## 3. Results

### 3.1. Basic Characteristics of the Participants

A total of 830 college students aged 18–22 years were investigated, with an average age of 19.57 ± 0.85 years. Males accounted for 29% (*n* = 241) and females accounted for 71% (*n* = 589). Sophomores accounted for 50.1% (*n* = 416) and Juniors 49.9% (*n* = 414). The proportion of participants with tinnitus, earache, and ear tightness was 51.6% (*n* = 428), 31.4% (*n* = 261), and 28.3% (*n* = 235), respectively. Overall, 61.2% of the students reported a family history of hearing loss. Grade distribution was statistically significantly different between the intervention group and the control group in male students (*p* < 0.001), while the grade distribution and average age were statistically significantly different in female students (*p* < 0.001) (Table 1).

### 3.2. Baseline Comparison and the Effects of Intervention on Hearing Health Knowledge and Belief

Before the intervention, there was no significant difference in the total scores of hearing health knowledge and health belief between the intervention group and control group in male and female students. Regarding the scores in the six sub-dimensions of the health belief, only the score for perceived barriers was significantly higher in the control group in female students (*p* = 0.045) (Table 2, Appendix A).

In female students, the scores of hearing health knowledge showed significant improvement in the intervention group at intervention cessation phase and 3 months after intervention cessation (T2–T1 IG 3.23 ± 5.80 vs. CG 1.45 ± 7.21, *p* = 0.007; T3–T1 IG 3.97 ± 8.04 vs. CG 1.95 ± 8.05; *p* < 0.014). Similarly, the scores of hearing health belief demonstrated significant improvement 3 months after intervention cessation (T3–T1 IG 1.32 ± 18.31 vs. CG −2.41 ± 15.98; *p* = 0.047). Among the sub-dimensions of hearing health belief, the scores of perceived severity and perceived self-efficacy showed improvement after intervention respectively (T2–T1 IG 1.15 ± 6.20 vs. CG −0.23 ± 6.49, *p* = 0.030; T2–T1 IG 0.28 ± 5.27 vs. CG −1.16 ± 5.42, *p* = 0.008) (Table 2).

In male students, the scores of hearing health knowledge and health belief demonstrated no significant improvement at the intervention cessation phase (T2) and 3 months after intervention cessation (T3) between both groups (Table 2).

### 3.3. Baseline Comparison and the Effects of Intervention on Hearing Health Related Behaviors

We focused on six types of hearing health behaviors related to the frequency of using headphones per day, duration of using headphones each time, using headphones at high volume, increasing headphone volume in noisy environments, using in-ear type headphones, and sleeping with headphones for listening to music or radio. Baseline analysis of the behaviors showed that the incidence of using headphones at high volume (≥40% of the total volume) was significantly higher in the intervention group in female students (*p* < 0.05) (Appendix A).

In female students, the intervention significantly reduced the proportion of students with the following four behaviors at intervention cessation phase and 3 months after intervention cessation, with the improvement in the average frequency of using headphones ≥3 times/day (T2–T1 IG vs. CG 1.913 (1.148–3.189), *p* = 0.013; T3–T1 IG vs. CG 1.612 (1.019–2.550), *p* = 0.041), the average duration of using headphones ≥30 min each time (T2–T1 IG vs. CG 1.540 (1.014–2.339), *p* = 0.043; T3–T1 IG vs. CG 1.854 (1.233–2.788), *p* = 0.003), volume of the headphones ≥40% of the total volume (T2–T1 IG vs. CG 2.416 (1.351–4.320), *p* = 0.003; T3–T1 IG vs. CG 2.224 (1.275–3.882), *p* = 0.005) and sleeping with headphones for listening to music or radio (T2–T1 IG vs. CG 1.725 (1.107–2.689), *p* = 0.016; T3–T1 IG vs. CG 1.991 (1.284–3.088), *p* = 0.002) (Table 3).

In male students, the intervention significantly reduced the proportion of students sleeping with headphones for listening to music or radio 3 months after intervention cessation (T3–T1 IG vs. CG 2.554 (1.184–5.509), *p* = 0.017) (Table 3).

## 4. Discussion

The health belief model is one of the earliest theories used to explain and predict health behavior [15], and includes six sub-dimensions: perceived susceptibility, perceived severity, perceived benefits, perceived barriers, self-efficacy, and cues to action [16]. Based on this model, we performed a 3-month hearing health intervention with a combination of online and offline educational activities and found that it successfully increased the hearing health knowledge and health belief and also reduced the incidence of hearing risk behaviors among the college students, especially in females.

In this study, we found that the 3-month health education intervention effectively improved the level of hearing-health knowledge in female students, with the good effect well-maintained 3 months after the intervention cessation. Similarly, Khan et al. conducted a six-week hearing intervention using computer training, classroom training, and classroom plus application (APP) training on high school students and found that all the three forms increased the average hearing health knowledge score by 20.0%, 14.2%, and 16.3%, respectively [17].

In female students, the effect of the intervention on the total scores of hearing health belief was detected only 3 months after the intervention cessation. Keppler et al. reported that a 6-month hearing education program improved participants’ attitudes and beliefs about hearing protector devices [13]. In contrast, Khan et al. claimed that the attitudes of 50 adolescent farmworkers toward hearing protection improved significantly after a 6-week hearing protection intervention [17]. We suggest that the effect of intervention probably depends on the intervention period and the characteristics of the participants. The consequences of hearing damage are not immediate if the college students fail to perceive the seriousness though they currently have risk behavioral habits, and a longer intervention is probably needed. The scores of perceived severity and self-efficacy showed improvement after intervention but weakened in the maintenance phase in female students. The perceived severity and self-efficacy are important for facilitating both engagement in health-promoting behaviors and maintenance of healthy habits [18], which remind us that persistent intervention is necessary for maintaining them at a high level.

The ultimate goal of improving the knowledge and belief in hearing protection is to reduce the incidence of hearing health risk behaviors. Common harmful behaviors leading to hearing loss include frequent attendance of concerts, karaoke sessions, high-noise places such as nightclubs, and using of personal music players [19,20,21]. Smart mobile devices have become necessary electronic products for college students, and headphones are indispensable auxiliary tools [8]. Previous studies have shown the need for an education program to promote healthy and safe listening habits among students [22,23]. Sunny et al. surveyed 388 students from the Medical School of the University of Lagos, Nigeria, and found that the usage rate of earphones was 95.6%, and plug-in earphones were the most common type of earphones [22]. Seedat et al. surveyed 269 first-year health science students at a university in South Africa. About 90.7% of the participants used personal listening devices, with 14.9% using them at a high volume and 52.7% reporting use of more than 2 h a day [23]. The present study focused on the headphones use behaviors. After the intervention, the incidences of most harmful behaviors among the students in the intervention group were significantly reduced after the intervention or in the maintenance phase, including the frequency of using headphones per day, duration of using headphones each time, and proportion of using headphones at high volume in female students, and reduced the behaviors of sleeping with headphones working in females and males.

This suggests that intervention programs can play a catalytic role. Liang et al. reported that most teenagers tend to raise the music volume in the presence of background noise in order to hear the music better [8]. Sunghwa et al. also found that 66.2% of 1009 college students from South Korea used earplugs or in-ear headphones with their personal listening devices, and the majority increased the volume of the headphone in a noisy environment, thus increasing the risk of hearing loss [7]. Probably health education intervention could be a good choice to improve these hearing risk behaviors.

In male students, the intervention showed no effectiveness in improving the scores of hearing health knowledge and health beliefs and was less effective in reducing hearing risk behaviors compared with female students. It is probably attributable to that female students have a higher level of health awareness of personal health care than males [24]. Therefore, better intervention improvement was achieved in females, and more effective measures need to be explored for male students.

In the maintenance period, intervention effectiveness still can be seen by the reduced incidence of hearing health risk behaviors, indicating the purpose of cultivating behavioral habits for long-term hearing protection was achieved. It can be a good reference provided for the formulation of health education programs in the future. Marlenga et al. conducted a 16-year follow-up study among 392 students who had participated in a 2–3-year hearing protection intervention and found an increase in the long-term use of hearing protection devices [25]. Similarly, a follow-up study by Martin et al. proved that a sustainable hearing health promotion program effectively promoted the participants’ relevant hearing health knowledge and improvement of hearing behaviors [26].

The present study showed several advantages. Firstly, due to the teachers’ participation and students’ cooperation, we found that hearing health education is conducive to ensuring effective participation of students and reducing the attrition rate as much as possible. Secondly, timely communication and feedback between the two sides made it easier for students to obtain hearing health knowledge and encouraged students to develop good hearing habits. There is a proposal to include hearing protection in the students’ curriculum. University organizations are actively exploring how to incorporate the teaching of healthy hearing habits to college students [27]. Klein et al. advocated that an audiology major should be added to the learning courses of students majoring in the oral cavity so as to enrich their knowledge and understanding of hearing loss [27]. Finally, the large sample size ensured the representativeness of the health education effect.

The present study has some limitations. The participants only comprised students from a medical college and may not be representative of students from other colleges and universities. Since medical college students need to communicate with patients and perform auscultation, they need to have good hearing. Additionally, we mainly focused on the behaviors on earphone and cellphone usage, and many other hearing health risk factors like behaviors of seeking medical treatment for ear diseases were not investigated. Follow-up studies should be extended to students of other majors and more hearing health behaviors. Finally, the number of males and females in the intervention and control groups was uneven, and we have controlled for the possible bias to the maximum extent.

## 5. Conclusions

In this study, a health intervention program based on the health belief model was conducted among college students. It effectively improved hearing health knowledge and hearing health beliefs and effectively reduced the incidence of hearing health risk behaviors in female students but weakly in male students. This study demonstrates the potential benefits of providing hearing health intervention classes to decrease the incidence of hearing health risk behaviors in universities. But specifically, intervention measures targeted to different genders should be carried out.

## Figures and Tables

**Table 1 ijerph-18-01560-t001:** College students’ demographic characteristics between intervention group and control group in males and females.

Characteristic	Total (*n* = 830)	Male		*p*	Female		*p*
		Intervention Group (*n* = 61)	Control Group (*n* = 180)		Intervention Group (*n* = 358)	Control Group (*n* = 231)	
Age, years	19.57 ± 0.85	19.62 ± 0.637	19.70 ± 0.838 ^a^	0.512	19.42 ± 0.759	19.66 ± 0.991	0.002
Grade				<0.001			<0.001
Sophomore	416 (50.1)	45 (73.8)	58 (32.2) ^b^		220 (40.3)	93 (61.5)	
Junior	414 (49.9)	16 (26.2)	122 (67.8)		138 (59.7)	138 (38.5)	
Tinnitus				0.734			0.179
No	402 (48.4)	30 (49.2)	84 (46.7)		183 (51.1)	105 (45.5)	
Yes	428 (51.6)	31 (50.8)	96 (53.3)		175 (48.9)	126 (54.5)	
Earache				0.996			0.359
No	569 (68.6)	42 (68.9)	124 (68.9)		250 (69.8)	153 (66.2)	
Yes	261 (31.4)	19 (31.1)	56 (31.1)		108 (30.2)	78 (33.8)	
Ear tightness				0.461			0.359
No	595 (71.7)	47 (77.0)	130 (72.2)		259 (72.3)	159 (68.8)	
Yes	235 (28.3)	14 (23.0)	50 (27.8)		99 (27.7)	72 (31.2)	
Number of people with hearing loss in the family				0.909			0.993
0	322 (38.8)	25 (41.0)	73 (40.6)		136 (38.0)	88 (38.1)	
1	290 (34.9)	23 (37.7)	64 (35.6)		124 (34.6)	79 (34.2)	
≥2	218 (26.3)	13 (21.3)	43 (23.9)		98 (27.4)	64 (27.7)	

Note: Age is shown as Mean ± SD (Standard Deviation), and other demographic characteristics are showed as *n* (%); ^a^
*t*-test; ^b^ chi-square test; Boldface indicates statistical significance (*p* < 0.05).

**Table 2 ijerph-18-01560-t002:** Comparison of hearing health knowledge and belief before and after intervention in male and female students.

Variable	Male		*p*	Female		*p*
	Control	Intervention		Control	Intervention	
	(*n* = 180)	(*n* = 61)		(*n* = 231)	(*n* = 358)	
Score of Hearing Health Knowledge						
Baseline	30.37 ± 6.24	29.33 ± 6.77		31.51 ± 5.17	31.01 ± 5.47	-
T2–T1	1.03 ± 9.09	3.48 ± 8.93	0.180	1.45 ± 7.21	3.23 ± 5.80	**0.007**
T3–T1	1.20 ± 11.04	1.95 ± 12.29	0.887	1.95 ± 8.05	3.97 ± 8.04	**0.014**
Score of Hearing Health Belief						
Baseline	164.57 ± 18.14	163.87 ± 16.13		171.08 ± 13.98	168.08 ± 14.67	-
T2–T1	−1.56 ± 20.09	2.48 ± 18.73	0.368	−2.68 ± 16.00	−0.10 ± 16.54	0.183
T3–T1	−0.88 ± 21.70	−2.28 ± 24.47	0.459	−2.41 ± 15.98	1.32 ± 18.31	**0.047**
Perceived Susceptibility						
Baseline	13.05 ± 3.50	13.77 ± 3.50		13.77 ± 3.30	13.41 ± 2.98	-
T2–T1	−0.13 ± 4.23	−0.49 ± 4.21	0.417	−0.64 ± 4.04	−0.54 ± 3.58	0.861
T3–T1	0.56 ± 3.82	0.15 ± 4.55	0.290	0.26 ± 3.59	0.14 ± 3.86	0.734
Perceived Severity						
Baseline	37.84 ± 4.86	37.67 ± 4.68		38.86 ± 4.42	38.11 ± 4.74	-
T2–T1	−0.88 ± 7.88	0.43 ± 5.56	0.259	−0.23 ± 6.49	1.15 ± 6.20	**0.030**
T3–T1	−1.04 ± 8.57	−1.48 ± 10.08	0.777	−0.11 ± 5.87	1.03 ± 6.47	0.065
Perceived Benefits						
Baseline	30.81 ± 4.53	30.11 ± 3.96		31.06 ± 4.05	30.65 ± 4.15	-
T2–T1	−0.22 ± 5.91	0.49 ± 4.14	0.724	0.06 ± 5.13	0.28 ± 5.02	0.677
T3–T1	−1.38 ± 6.99	−1.67 ± 7.50	0.719	−0.50 ± 5.22	−0.21 ± 5.85	0.578
Perceived Barriers						
Baseline	24.92 ± 6.83	23.93 ± 6.62	-	25.84 ± 5.97	24.73 ± 5.79	-
T2–T1	−0.73 ± 8.60	0.39 ± 8.33	0.442	−0.87 ± 7.54	−0.81 ± 7.36	0.841
T3–T1	−1.74 ± 8.91	−1.67 ± 9.49	0.696	−3.20 ± 7.25	−1.48 ± 7.88	0.059
Cues to Action						
Baseline	28.69 ± 6.03	28.56 ± 4.85	-	30.11 ± 4.54	30.48 ± 4.30	-
T2–T1	0.24 ± 7.83	1.03 ± 5.48	0.759	0.15 ± 5.95	−0.47 ± 6.17	0.227
T3–T1	1.18 ± 7.06	1.39 ± 7.90	0.415	0.64 ± 5.94	0.36 ± 6.15	0.680
Perceived Self-Efficacy						
Baseline	29.26 ± 5.34	29.82 ± 5.08		31.43 ± 4.63	30.70 ± 4.58	-
T2–T1	0.17 ± 5.49	0.62 ± 6.01	0.705	−1.16 ± 5.42	0.28 ± 5.27	**0.008**
T3–T1	1.54 ± 6.67	1.00 ± 6.97	0.467	0.51 ± 5.13	1.49 ± 5.36	0.074

Note: All values are mean ± SD (Standard Deviation). T2–T1: The value obtained at T2 minus the baseline value (T1); T3–T1: The value obtained at T3 minus the baseline value (T1). Significant differences in intervention outcomes between groups were determined by using Repeated Measures ANOVA with age and grades being adjusted. Boldface indicates statistical significance (*p* < 0.05).

**Table 3 ijerph-18-01560-t003:** Binary logistic regression analysis of the efficiency of intervention on the hearing health related behaviors in college students.

Behaviors	Males (*n* = 241)				Females (*n* = 589)			
	T2–T1		T3–T1		T2–T1		T3–T1	
	OR (95% CI)	*p*	OR (95% CI)	*p*	OR (95% CI)	*p*	OR (95% CI)	*p*
The average frequency of using headphones ≥3 times/day	1.393 (0.654–2.964)	0.39	1.387 (0.672–2.861)	0.376	1.913 (1.148–3.189)	0.013	1.612 (1.019–2.550)	0.041
Average duration of using headphones ≥30 min each time	1.039 (0.507–2.127)	0.918	1.196 (0.604–2.368)	0.608	1.540 (1.014–2.339)	0.043	1.854 (1.233–2.788)	0.003
Volume of the headphones ≥40% of the total volume	1.948 (0.807–4.703)	0.138	1.219 (0.520–2.857)	0.649	2.416 (1.351–4.320)	0.003	2.224 (1.275–3.882)	0.005
Sleeping with headphones for listening to music or radio	1.605 (0.759–3.392)	0.216	2.554 (1.184–5.509)	0.017	1.725 (1.107–2.689)	0.016	1.991 (1.284–3.088)	0.002
Probability of increasing headphone volume in noisy environment ≥50%	1.014 (0.458–2.243)	0.972	1.037 (0.496–2.169)	0.923	1.032 (0.676–1.576)	0.884	1.453 (0.967–2.184)	0.072
Using in-ear type headphones	0.884 (0.333–2.348)	0.805	1.062 (0.408–2.763)	0.902	1.065 (0.626–1.812)	0.816	1.386 (0.812–2.364)	0.231

Note: T2–T1: Behavioral improvement from pre-intervention (T1) to the intervention cessation phase (T2) in the intervention group compared with that of the control group; T3–T1: Behavioral improvement from pre-intervention (T1) to 3 months after the intervention cessation (T3) in the intervention group compared with that in the control group; Boldface indicates statistical significance (*p* < 0.05).

## Data Availability

The data presented in this study are available on request from the corresponding author. The data are not publicly available due to privacy or ethics.

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
