# Peer review of "Health Education Intervention on Hearing Health Risk Behaviors in College Students"

_ijerph, 2021, doi:10.3390/ijerph18041560_

Round 1

Reviewer 1 Report

In the present work the authors present an educational intervention to avoid hearing risks in students, these are some of my comments:

Identifying a baseline significance between women and sofomors in both groups is rather a selection bias that should be controlled, the same for the use of headphones in volume.

How did the authors check the increase in fruit intake?

Who carried out the application of the intervention questionnaire?

The main concern of the study is that no novel data is presented. It is quite predictable that the education of medical students will improve their habits in most cases.

Author Response

Dear reviewer

Thank you very much! We appreciated the important comments about our paper submitted to International Journal of Environmental Research and Public Health (ijerph-1066237: Health education intervention on hearing health risk behaviors in college students). We have revised the manuscript in accordance with your and reviewers’ constructive comments, and carefully proof-read the manuscript. The response to your comments are as follows.

Comment 1: Identifying a baseline significance between women and sophomores in both groups is rather a selection bias that should be controlled, the same for the use of headphones in volume.

Response 1: Thank you very much for your valuable advice. Previous data analysis results did show a selection bias. In the revised version, we divided the total study population by gender into male students and female students. The effect of health education intervention on students’ hearing health knowledge, health belief and health related behaviors were analyzed respectively in different gender subgroups. Considering that there were slight differences in grades distribution and age between the intervention group and control group in male students or female students. We have adjusted the age and grades in the statistical analysis so as to control the influence of confounding factors on the results to the greatest extent.

Comment 2: How did the authors check the increase in fruit intake?

Response 2: Thank you for your question. The vegetable and fruit intake survey was conducted in accordance with the guidelines from Chinese Dietary Guidelines 2016. The investigators surveyed the participants by face to face to instruct them to recall their daily vegetable and fruit intake. In this study, we reflect the rise and fall of vegetable and fruit intake based on the change in the proportion of students who ate more than 500g of fruits and vegetables per day, rather than directly measuring whether the weight of fruits and vegetables intake for each student was increased or not.

Comment 3: Who carried out the application of the intervention questionnaire?

Response 3: Thank you for you reminding. The questionnaire survey was conducted by investigators including teachers and graduate students who received strictly professional training to ensure data collection quality. They were able to provide an accurate interpretation during questionnaire collection by face to face interviews, in order to minimize the subjective bias of the participants. If some information in the questionnaire was missing, the investigators checked with the participants in order to complete the questionnaires. We have added this information in the paper.

Comment 4: The main concern of the study is that no novel data is presented. It is quite predictable that the education of medical students will improve their habits in most cases.

Response 4: Thank you very much for your pertinent criticism! Previous results before revision may not be very attractive and innovative. But we have revised the data analysis according to your wonderful suggestions. After gender stratification analysis, we found that the health education intervention had a better intervention effect on females students’’ hearing health knowledge, health belief and health related behaviors, while the intervention effect for male students was weak. The findings serve as a reminder that health interventions for men may need to be better designed, both in terms of timing and approaches.

Reviewer 2 Report

Health and hearing is a topic which brought many interesting studies. Since WHO set up WOrld Hearing Forum a lot interest is focused eg. for healthy behaviour . That study is focused on student group which are not equal (male / female) . I can not find bioethical committee approval . There is no clear definition about tinnitus , earache etc. Regarding tool - questionnaire tere is not information about validation . Aspects which are mentioned there are too superficial. Figures should be presented with more details . There is good and sufficient discussion but regarding issues mentioned above there is high need to improve that manuscript and I recommend to resubmit .

Author Response

Dear reviewer

Thank you very much! We appreciated the important comments about our paper submitted to International Journal of Environmental Research and Public Health (ijerph-1066237: Health education intervention on hearing health risk behaviors in college students). We have revised the manuscript in accordance with your constructive comments, and carefully proof-read the manuscript. The response to your comments are as follows.

Comment 1: That study is focused on student group which are not equal (male /female) .

Response 1: Thank you very much for your valuable advice. Previous data analysis results did show a selection bias. In the revised version, we divided the total study population by gender into male students and female students. The effect of health education intervention on students’ hearing health knowledge, health belief and health related behaviors were analyzed respectively in different gender subgroups. Considering that there were slight differences in grades distribution and age between the intervention group and control group in male students or female students. We have adjusted the age and grades in the statistical analysis so as to control the influence of confounding factors on the results to the greatest extent.

Comment 2: I can not find bioethical committee approval. 

Response 2: Thank you for your reminding! Bioethical committee approval is located at the end of this paper as required by the guideline of the journal. It is as follows. “Institutional Review Board Statement: The study was conducted according to the guidelines of the Declaration of Helsinki, and ap-proved by the Institutional Review Board of Hangzhou Nor-mal University Ethics Committee (No.2017LL107).”

Comment 3: There is no clear definition about tinnitus, earache etc.

Response 3: Thank you for your reminding! Tinnitus was divided into two conditions: no tinnitus within the past year, and occasional tinnitus (less than or equal to once a week) or frequent tinnitus (more than or equal to twice a week). Earache and ear tightness were similarly classified. We have added this section into the methods.

Comment 4: Regarding tool - questionnaire here is not information about validation. Aspects which are mentioned there are too superficial.

Response 4: Thank you for your reminding! The questionnaire was validated in previous research, which include 2088 participants. The questionnaire was validated in previous research with 2088 participants involved. Its Cronbach’s coefficient was 0.888, and the KMO value was 0.872. Principal component analysis showed that confirmatory factors accounted for 64.395% of the total variance.  

Comment 5: Figures should be presented with more details.

Response 5: Thank you for your suggestion! We have deleted the original figure, and the data was presented using tables.

Comment 6: There is good and sufficient discussion but regarding issues mentioned above there is high need to improve that manuscript and I recommend to resubmit. 

Response 6: Thank you for your suggestion! We have made careful modifications according to your valuable suggestions to improve the quality of this paper.

Reviewer 3 Report

I want to congratulate the authors on this way of presenting their study.

However, several points as indicated below need to be addressed by authors to improve the quality of the article:

  1. In the abstract you mention “studies on their hearing risk behaviors are inadequate” (11-12). Can you better explain and substantiate this sentence?

  1. In the abstract you mention “After the intervention, the score of hearing health-related knowledge in the intervention group was significantly increased” (18-19). The presentation of the results in this part of manuscript should be improved, with reference to the value of statistical significance (p= ?)

  1. Does the control group have the same sociodemographic characteristics as the intervention group? You only mention age. Other factors, such as hearing and the habits of young people in relation to noise exposure, must be considered beforehand so that the two groups (control and intervention) are similar in terms of audiological and sociodemographic characteristics (69-73). At the front you present Table 1 with the sociodemographic characteristics of the two groups, but shouldn't you refer to these characteristics for the constitution of the two groups (intervention and control) in this part of the text? It seems that this balance parameter between the groups was not achieved, as the results refer: “In comparison with the control group, the proportions of women and second year students are significantly higher in the intervention group (p <0.001)” (125-126). This aspect should be considered as a study limit.

  1. The inclusion and exclusion criteria must be described in the Materials and Methods.

  1. In the Materials and Methods there is no reference to ethical issues, namely approval by an ethics committee or the signed an informed consent form before the experiment.

  1. Are the questionnaires used validated? How can we access this questionnaire? (81-96). Despite the characterization of the questionnaires during the Materials and Methods it’s important the validation of the instruments used in the research.

  1. In the manuscript, you consider nine types of behaviors related to hearing health. I would like to know why you considered the relationship between “consumption of vegetables and fruits <500 g / day”. (105-106) and behaviors related to hearing health? It should be clearer to use this specific topic in in the Materials and Methods. It is only later in the discussion (lines 248 -256), that this question is founded.

  1. In the results you mentioned “As for the nine types of hearing health-related behaviors, the difference was significant only for the rates of volume of the headphones ≥40% of the total volume and sleeping with headphones for listening to music or radio (p<0.05) (Table 2). (136-142). How do you reach this result using table 2?

  1. Table 3 should be formatted again?

Author Response

Dear reviewer

Thank you very much! We appreciated the important comments about our paper submitted to International Journal of Environmental Research and Public Health (ijerph-1066237: Health education intervention on hearing health risk behaviors in college students). We have revised the manuscript in accordance with your constructive comments, and carefully proof-read the manuscript. The response to your comments are as follows.

Comment 1: In the abstract you mention “studies on their hearing risk behaviors are inadequate” (11-12). Can you better explain and substantiate this sentence?

Response 1: Thank you for your suggestions! We used this sentence to mean that hearing intervention on behavioral interventional studies on college students are inadequate, because when we did the literature review, we found that there really wasn't a lot of intervention research on hearing loss risk behaviors in the population of college students.

Comment 2: In the abstract you mention “After the intervention, the score of hearing health-related knowledge in the intervention group was significantly increased” (18-19). The presentation of the results in this part of manuscript should be improved, with reference to the value of statistical significance (p= ?)

Response 2: Thank you for your suggestions!  We have reanalyzed the data by gender stratification. In female students, the scores of hearing health knowledge showed significant improvement in the intervention group at intervention cessation phase and 3 months after intervention cessation (T2-T1 IG 3.23±5.80 vs.CG 1.45±7.21, p=0.007; T3-T1 IG 3.97±8.04 vs. CG 1.95±8.05; p<0.014). In male students, the scores of hearing health knowledge demonstrated no significant improvement after the intervention and during the maintenance phase (Detailed information in Table 2 in the revised paper).

Comment 3: Does the control group have the same sociodemographic characteristics as the intervention group? You only mention age. Other factors, such as hearing and the habits of young people in relation to noise exposure, must be considered beforehand so that the two groups (control and intervention) are similar in terms of audiological and sociodemographic characteristics (69-73). At the front you present Table 1 with the sociodemographic characteristics of the two groups, but shouldn't you refer to these characteristics for the constitution of the two groups (intervention and control) in this part of the text? It seems that this balance parameter between the groups was not achieved, as the results refer: “In comparison with the control group, the proportions of women and second year students are significantly higher in the intervention group (p <0.001)” (125-126). This aspect should be considered as a study limit.

Response 3: Thank you for your suggestions! In the present study, the sociodemographic characteristics of the students include the age, grades, tinnitus, ear ache, ear tightness, and number of people with hearing loss in the family. Actually, the habits of young people in relation to noise exposure was not collected, and the students who were diagnosed as hearing loss were excluded from our research in order to control the bias.

Previous data analysis results did show a selection bias. In the revised version, we divided the total study population by gender into male students and female students. The effect of health education intervention on students’ hearing health knowledge, health belief and health related behaviors were analyzed respectively in different gender subgroups. Considering that there were slight differences in grades distribution and age between the intervention group and control group in male students or female students. We have adjusted the age and grades in the statistical analysis so as to control the influence of confounding factors on the results to the greatest extent.

Comment 4: The inclusion and exclusion criteria must be described in the Materials and Methods.

Response 4: Thank you for you reminding! The inclusion criteria was as follows: 1) medical students living on the university campus; 2) signed the informed consent and agreed to participate in the intervention program; and 3) without hearing loss. The hearing status of these college students were tested in our previous research.  We have added this information in the Materials and Methods.

Comment 5: In the Materials and Methods there is no reference to ethical issues, namely approval by an ethics committee or the signed an informed consent form before the experiment.

Response 5: Thank you for your reminding! Bioethical committee approval is located at the end of this paper as required by the guideline of the journal. It is as follows. “Institutional Review Board Statement: The study was conducted according to the guidelines of the Declaration of Helsinki, and ap-proved by the Institutional Review Board of Hangzhou Nor-mal University Ethics Committee (No.2017LL107).”

Comment 6: Are the questionnaires used validated? How can we access this questionnaire? (81-96). Despite the characterization of the questionnaires during the Materials and Methods it’s important the validation of the instruments used in the research.

Response 6: Thank you for your reminding! The questionnaire was validated in previous research, which include 2088 participants. The questionnaire was validated in previous research with 2088 participants involved. Its Cronbach’s coefficient was 0.888, and the KMO value was 0.872. Principal component analysis showed that confirmatory factors accounted for 64.395% of the total variance.

Comment 7: In the manuscript, you consider nine types of behaviors related to hearing health. I would like to know why you considered the relationship between “consumption of vegetables and fruits <500 g / day”. (105-106) and behaviors related to hearing health? It should be clearer to use this specific topic in in the Materials and Methods. It is only later in the discussion (lines 248 -256), that this question is founded.

Response 7: Thank you for your suggestion! In our previous researches, we found high level of daily consumption of vegetables and fruits was a protective factor of hearing health, which was also evidenced by other literatures. Therefore, we investigated the students’ behavior of daily consumption of vegetables and fruits in this study. After the major revision, we have deleted the information about consumption of vegetables and fruits to make the investigated behaviors focus on earphones and cellphone usage.

Comment 8: In the results you mentioned “As for the nine types of hearing health-related behaviors, the difference was significant only for the rates of volume of the headphones ≥40% of the total volume and sleeping with headphones for listening to music or radio (p<0.05) (Table 2). (136-142). How do you reach this result using table 2?

Response 8: Thank you for your suggestion! We have reanalyzed the data according you and other reviewers’ comments. Binary logistic regression was used to estimate the efficiency of health intervention on the hearing health related behaviors. If one type of behavior was found to have improved at T2 or T3, compared to T1, the intervention was considered effective(Y=1), whereas if there was no improvement or the behaviors were found to have worsened, the intervention was considered ineffective(Y=0). The detailed information was in Table 3.

Comment 9: Table 3 should be formatted again?

Response 9: Thank you for your suggestion! We have reanalyzed the data according to the included comments of reviewers, and presented it in a new table.

Round 2

Reviewer 1 Report

No comments

Author Response

Dear reviewer

Thank you very much! We appreciated your approval of our revised manuscript based on your valuable comments and suggestions. (ijerph-1066237: Health education intervention on hearing health risk behaviors in college students).

Best Wishes for You!

Sincerely,

Liangwen Xu

Reviewer 2 Report

Dear Authors

I checked the manuscript as well as remarks from previous reviews. There is improvement no doubt . Especially in technical issues. I found method (even after change and more information about validation ) not enough sufficient for such journal and if there is chance to put additional questionnaire or evaluation form which could more familiar for journal recipients to recognize . 

Author Response

Dear reviewer,

Thank you very much! We appreciated the important comments about our paper submitted to International Journal of Environmental Research and Public Health (ijerph-1066237: Health education intervention on hearing health risk behaviors in college students). The response to your comments are as follows.

Best Wishes for You!

Sincerely,

Liangwen Xu

Comment 1: I found method (even after change and more information about validation ) not enough sufficient for such journal and if there is chance to put additional questionnaire or evaluation form which could more familiar for journal recipients to recognize.

Response 1Thank you very much for your suggestion. It should be much better if we added much more information about the questionnaire. The reason why we did not put much more information about the questionnaire is that we have another paper on this questionnaire (titled “Construction and evaluation of hearing health belief scale for medical students”), which has been submitted to a Chinese journal and is currently under review. Thank you very much for your consideration.